# Evaluation of the Properties of 3D-Printed Ti Alloy Plates: *In Vivo* and *In Vitro* Comparative Experimental Study

**DOI:** 10.3390/jcm12020444

**Published:** 2023-01-05

**Authors:** Qi Wang, Wael Telha, Yange Wu, Bassam Abotaleb, Nan Jiang, Songsong Zhu

**Affiliations:** 1State Key Laboratory of Oral Diseases and National Clinical Research Center for Oral Diseases, West China Hospital of Stomatology, Sichuan University, Chengdu 610041, China; 2Department of Orthognathic and TMJ Surgery, West China Hospital of Stomatology, Sichuan University, Chengdu 610041, China; 3Department of Orthodontics, West China Hospital of Stomatology, Sichuan University, Chengdu 610041, China

**Keywords:** 3D printing, Ti6Al4V, Ti plates, selective laser melting, biocompatibility, morphological properties, rigid internal fixation, bone–plate contact rate

## Abstract

Titanium (Ti)-based implants play a significant role in rigid internal fixation in maxillofacial surgery. No study has reported that three-dimensional-printed Ti alloy plates (3D-Ti plates) have comprehensively excellent properties similar to standard plates (Matrix-MANDIBLE, SYNTHES, Switzerland) (Synthes-Ti plates). In this work, we manufactured 3D-Ti plates by selective laser melting with Ti6Al4V powder. The surface morphology, mechanical properties, and bone–plate contact rate of the 3D-Ti plates and the Synthes-Ti plates were characterized and compared via electron microscopy, atomic force microscopy, Vickers hardness test, three-point bending test, and software calculation. Human bone marrow stromal cells (HBMSCs) were cultured on the plates to test their biocompatibility. Importantly, the 3D-Ti plates were placed into a mandibular fracture model to assess the effect of medical application for 4 and 24 weeks. The 3D-Ti plates were demonstrated to have similar biocompatibility and stability for rigid internal fixation with the Synthes-Ti plates, lower roughness (106.44 ± 78.35 nm), better mechanical strength (370.78 ± 1.25 HV10), and a higher bone–plate contact rate (96.9%). These promising results indicate the feasibility of using 3D-Ti plates for irregular shapes and complex anatomical structures in a clinical context.

## 1. Introduction

Ti-based implants have been widely used in orthopedics and dental surgery due to their excellent mechanical properties, chemical stability, and biocompatibility [1]. Titanium 6-aluminium 4-vanadium alloy (Ti6Al4V alloy), recognized as the most popular Ti alloy, is essential for rigid internal fixation in reparative, corrective, and reconstructive surgery [2]. However, the plates must be shaped to fit the bone surface due to the irregular shape and complex anatomical structure of the maxillofacial region. Clinically, despite repeated attempts at plate shaping, the scale needs to be satisfactorily matched, which can adversely affect primary bone-fracture consolidation [2,3]. Moreover, plate shaping can generate micro fissures, which result in local stress concentration or even the failure of rigid internal fixation [3,4]. The literature shows that plate fractures occur in 2.9–10.7% of mandibular injury cases [5]. Loukota and Shelton reported that more than five times bending the plates would reduce the rigidity of the flexion part [6]. The greater the Ti plate bending range, the greater its maximum stress, which would deviate more from the design scheme [7]. Obtaining the perfect shape and attachment of the Ti alloy plates is still a vital issue in oral and maxillofacial surgery.

In order to address this issue, three-dimensional (3D) printed plates were introduced into the field of orthopedic and maxillofacial surgery. For example, Jardini et al. designed and fabricated a 3D-printed customized implant for the surgical reconstruction of a large cranial defect [8]. Philippe et al. utilized integral 3D-printed individual Ti plates to achieve intraoperative positioning and fixation of the bone in maxillary segmental osteotomy [9]. Despite some progress, the safety and mechanical stability of 3D-printed Ti plates still need to be determined, which limits their large-scale clinical application. Until now, no deep study has been performed for the rigid internal fixation application of 3D-printed Ti6Al4V alloy plates in the maxillofacial region.

In this study, we deployed the 3D-printed Ti6Al4V alloy plates (3D-Ti plates) manufactured by selective laser melting (SLM) using Ti6Al4V powder and compared their mechanical properties and shaping accuracy with standard Ti alloy plates (Synthes-Ti plates, Matrix-MANDIBLE, 04.503.750, SYNTHES, Zuchwil, Switzerland). The 3-Matic Research (Materialise NV, Leuven, Belgium) software calculated the bone–plate contact rate to evaluate the accuracy of the plate shaping. Human bone marrow stromal cells (HBMSCs) were cultured on the two groups of Ti plates for cytocompatibility testing. A mandibular fracture model was conducted on rabbits to explore the feasibility of the 3D-Ti plate fixation under the functional mechanics and physiological movement of the maxillofacial structure.

## 2. Materials and Methods

### 2.1. Preparation of the Ti Plates Specimens

The typical commercial four-hole Synthes-Ti plates were chosen as the control group. We used the Micro-CT to scan the dimension and details of the control group. The digital model of the commercial plate was imported to Concept Laser (Lichtenfels, German), and the 3D-Ti plates were created by SLM using Ti6Al4V powder (Tai Fu Mei^TM^, Chengdu, China).

### 2.2. Surface Morphology

The surfaces of the Synthes-Ti plates and the 3D-Ti plates were scanned by Atomic force microscopy (AFM, Shimadzu SPM-9700, Kyoto, Japan), and the average surface roughness (Sa), vertical range (Sz), and root-mean-square surface roughness (Sq) were compared. Six points were randomly scanned on each plate.

The Synthes-Ti and 3D-Ti plate surfaces were magnified 300 times and 1000 times using scanning electron microscopy (SEM, FEI, Eindhoven, The Netherlands). Moreover, the Synthes-Ti plates were shaped by simulating clinical use and observed by SEM scanning again.

### 2.3. Mechanical Properties

The Ti plates’ hardness was assessed using the Vickers Hardness Test by a Vickers Indenter (HVST-1000Z, Hong Kong, China) with 10 kg of force. Three plates were randomly measured in each group, and three points were randomly selected for each plate.

The three-point bending test was carried out on the universal mechanical testing machine (Instron 3360, Norwood, MA, USA) three times for each group. The span’s distance was 8 mm, and the roller bars were 3 mm in diameter. The bars contacted the middle rectangular part of the four-hole plates, which were 30 mm in length, 5 mm in width and 1 mm in thickness. At room temperature (20 °C), the displacement-controlled test was conducted at 1 mm/min until the sample fell off or broke. The sensors collected the load and displacement data automatically, and the ultimate stress and elasticity modulus were subsequently calculated.

### 2.4. In Vitro Cell Behavior

The HBMSCs were purchased from Oricell^®^ (Cyagen Biosciences, Guangzhou, China). Cells in passages 2 to 4 were used in this study. The cell behavior on the 3D-Ti plates (experimental group) was compared with the Synthes-Ti plates (control group). The samples used in cell experiments were 5 × 5 mm plates cut from original Ti plates.

#### 2.4.1. Cell Adhesion

HBMSCs were seeded, respectively, on the sterilized samples at a concentration of approximately 2 × 10^4^ cells. After incubation for 2 h and 6 h, the samples were thoroughly rinsed to remove the unattached cells, fixed, and dehydrated. The cytoskeleton was fluorescently labeled by rhodamine phalloidine (Sigma, St. Louis, MO, USA) and observed under a laser scanning confocal microscope (CLSM, Olympus, Tokyo, Japan). The quantitative analysis was conducted by Image J. SEM also observed the cell morphology.

#### 2.4.2. Cell Proliferation

HBMSCs were seeded on the samples at the same concentration as the cell adhesion experiment. After 1, 3, 5, and 7 days, Cell Counting Kit-8 (cck-8, APExBIO, Houston, TX, USA) was added and incubated at 37 °C for 2.5 h. The cell viability was evaluated by the absorbance value determined at 450 nm.

### 2.5. Animal Study

This study was approved by the Ethics Committee of the Key Laboratory of Oral Diseases and West China School of Stomatology, Sichuan University (Approval number: WCHSIRB-D-2020-251). A total of 24 New Zealand male white rabbits weighing 2.5 ± 0.3 kg, aged 6–8 months, were randomly assigned into two groups (12 rabbits in each group). Animal care requirements set by the state’s Animal Research Committee were followed.

#### 2.5.1. Accuracy Evaluation of the Ti Plate Shaping

The data from rabbit skulls were obtained from spiral-CT, converted to STL, imported to 3-Matic Research 11.0, and reconstructed. The 3D-Ti plates were designed at specific positions to be adapted for the bone surfaces (maxilla, mandible, condyle, and zygomatic bone). The skull model was printed using resin 3D printing, while the designed 3D-Ti plates were fabricated using SLM. For the control group, five skilled doctors bent the Synthes-Ti plates to ensure maximum contact with the same surfaces of resin models. The two groups of Ti plates were mounted on the resin models and scanned by Carestream Intraoral Scanner cs3600 (Carestream Health, Inc., Shanghai, China). The data were loaded into the 3-Matic Research to compare the surface areas of both Ti plates and the greatest contact areas between the Ti plates and the bone surfaces.

#### 2.5.2. Animal Model and Surgery

All the surgical interventions were performed in a qualified veterinary operating room. General anesthesia was induced by the intramuscular injection of 30 mg/kg of ketamine and 2 mg/kg of diazepam. After skin preparation, disinfection, and draping, an additional local anesthetic was added. A 2 cm incision was made along the inferior margin of the mandible. The fascia and muscle were bluntly dissected layer by layer until the mandible was exposed.

Firstly, we designed the ideal vertical fracture line from the distal end of the last molar to the inferior margin of the mandible. The four-hole Ti plates were mounted on the mandibular surface perpendicular to this fracture line, and four screw holes, two on each side, were drilled with sterile water cooling. The Ti plates were put aside temporarily, and the mandible was sawed off according to the designed line under cooling irrigation. Four Ti nails (Matrix-MIDFACE, 04.503.406.01c, SYNTHES, Zuchwil, Switzerland) were screwed into the pre-drilled holes to fix the plates. Twelve rabbits were implanted with the 3D-Ti plates, while the other 12 rabbits were implanted with the Synthes-Ti plates. Non-resorbable sutures were then used to sew up the wound. The animals received injectable antibiotics for 5 days after surgery, and sutures were removed after 7 days. Twelve rabbits were sacrificed in the 4th and 24th week, respectively, with 6 rabbits in each group.

#### 2.5.3. Biosafety Assessment

The whole blood and serum samples were collected in the 2nd, 4th, and 24th weeks, respectively. A blood routine was performed to examine hemoglobin concentration (HGB), mean corpuscular volume (MCV), mean red blood cell hemoglobin volume (MCH), white blood cell counts (WBC), red blood cell counts (RBC), hematocrit (HCT), red blood cell distribution width (RDW), and mean platelet volume (MPV). Blood biochemistry was performed to examine alanine aminotransferase (ALT), albumin (ALB), alkaline phosphatase (ALP), total bilirubin (TBIL), and creatinine (CRAE).

Histological analysis of the visceral organ tissues was conducted in the 4th and 24th week, including the heart, liver, spleen, lung, kidney, and brain. The tissue and organ samples were fixed, sectioned, stained with hematoxylin and eosin (H&E), and observed under a microscope.

#### 2.5.4. Evaluation Effectiveness of the Ti Plates in the Fracture Fixation

The effectiveness of the Ti plates was evaluated by assessing the fracture healing. The mandibles were separated after the rabbits were sacrificed and underwent an X-ray examination. Furthermore, the healed part of the fracture, about 10 × 10 × 8 mm cubic bone, was cut from the mandible and fixed in formalin for further analysis. The samples were analyzed by a Micro-CT scanner system (filter Al 0.5 mm, 70 kV, 200 µA, 15.0 μm), defining the most central 5 × 5 mm square region of the cubic bone with 200 slices as the volume of interest (VOI). Data analysis was performed by the SCANCO Evaluation application, including the values of bone volume per total volume (BV/TV), trabecular number (Tb. N), trabecular thickness (Tb. Th), and trabecular separation (Tb. Sp). Then, the scanned samples were decalcified, dehydrated, embedded, sectioned, stained with H&E and Masson, and observed under a microscope.

#### 2.5.5. Evaluation of the Mechanical Stability of the Ti Plates

After the sacrifice of the rabbits in the 24th week, the Ti plates of both groups were removed, and the three-point bending test was conducted with the same method mentioned before. The Ti plates’ ultimate stress and elasticity modulus were compared between different groups and within groups (original and postoperative).

### 2.6. Statistics

Data are expressed as the mean and standard deviation (SD). Independent sample *t*-tests and paired sample *t*-tests were applied for the statistical analysis using SPSS 22.0 (SPSS, Chicago, IL, USA). The significance level was set as a two-tailed 0.05.

## 3. Results

### 3.1. The Mechanical Properties and Surface Morphology of the Ti Plates

The hardness was 370.78 ± 1.25 HV10 for the 3D-Ti plates and 156.04 ± 1.37 HV10 for the Synthes-Ti plates (Figure 1b shows images in indentations). The hardness of the 3D-Ti plates was higher than that of the Synthes-Ti plates, with significant statistical differences (*p* < 0.0001) (Table 1). Under the three-point bending test analysis, it was observed that both the ultimate stress and the elasticity modulus of the 3D-Ti plates were more robust than those of the Synthes-Ti plates, and the differences were statistically significant (*p* < 0.0001) (Figure 1a, Table 1). Hence, the 3D-Ti plates had better extreme pressure resistance and were more difficult to deform.

Figure 1c shows AFM images of the Ti plate surfaces. The surface of the Synthes-Ti plates had tiny depressions, while the surface of the 3D-Ti plates had constant strip rises and falls. The surface of all plates had good roughness (Figure 1d). The Sa of the 3D-Ti plates was 106.44 nm, much lower than the 209.46 nm of the Synthes-Ti plates (*p* < 0.05).

Figure 2a shows the neat and smooth surface of the Synthes-Ti plates, while the surface of the 3D-Ti plate shows parallel scratches. Once the Synthes-Ti plates were shaped, rough marks would emerge on their surface. SEM images revealed the details of the different surface morphology. The surface of the Synthes-Ti plates showed shallow pits. The 3D-Ti plates’ surfaces were flat and smooth, and polishing marks in the same direction could be seen. Nevertheless, the shaped Synthes-Ti plates surface showed more manifest pits and micro-cracks (Figure 2b).

### 3.2. In Vitro Cell Behavior

#### 3.2.1. Cell Adhesion

The CLSM and SEM images in Figure 3a demonstrate the morphology and distribution of BMSCs. At 2 h, BMSCs on both the Ti plates were widely dispersed and simultaneously had many filopodia extensions. At 6 h, BMSCs on the surfaces of the two plates had increased cell adhesion and were densely linked. The cell fluorescence area of the 3D-Ti plates group was the same as that in the Synthes-Ti plates at both 2 h and 6 h, according to the red fluorescence-stained cytoskeleton (Figure 3b) (*p* > 0.05). The cell fluorescence area of the two groups increased at 6 h compared with 2 h, but the difference was not statistically significant (*p* > 0.05).

#### 3.2.2. Cell Proliferation

Cell proliferation on the Ti plate surfaces was determined by a cck-8 assay. As shown in Figure 3c, cell reproductive capacity increased on the two sample surfaces from 1 d to 7 d, and there was no statistical difference between the groups at 1 d, 3 d, and 5 d (*p* > 0.05). Cell reproductive capacity after 7 days of culture was higher on the surface of the 3D-Ti plate than that of the Synthes-Ti plates (*p* < 0.05). These results suggest that the cytocompatibility of the 3D-Ti plates is no less than that of the Synthes-Ti plates.

### 3.3. In Vivo Animal Study

#### 3.3.1. Accuracy of the Ti Plate Shaping

The 3D-Ti plates were successfully designed on the rabbit head model: on the maxilla, mandible, zygoma, and condyle (Figure 4a). The 3D-Ti plates fit perfectly to the specified model surface and could accurately remain in the designed position without slipping (Figure 4b). The Synthes-Ti plates, bent by skilled doctors, could also be placed steadily on the model surface (Figure 4a,b). The ratio of the surface area of the Ti plate in contact with the underlying bone to the plate surface area was defined as the bone–plate contact rate. The average bone–plate contact rate of 3D-Ti plates was 96.9%, 30.9% more than that of the Synthes-Ti plates. The highest bone–plate contact rate of the 3D-Ti plates was 98.1% among the four positions. The rates of the Synthes-Ti plates were 73.6% and 70.2% in the mandibular and maxillary positions but only 62.6% in the condyle. In the depressed zygomatic position, the rate was barely 57.6% (Figure 4c).

#### 3.3.2. Biosafety of the Ti Plates

Based on the H&E staining images of the animals’ viscera (Figure 5a), the morphological structures of all tissues were normal and typical, with no significant difference between the two groups at either 4 or 24 weeks after surgery, which was corroborated by the blood routine and blood biochemistry results (Figure 5b). In the 2nd week, the MCH, RDW, and ALT values of the 3D-Ti plates group were higher than those of the control group (*p* < 0.05, *p* < 0.05, *p* < 0.05, separately), but the differences disappeared in the 4th and 24th week. The values of ALP and CREA in the 3D-Ti plates group were higher in the 4th week than in the experimental group (*p* < 0.05), but there was no difference in the 24th week. There was no statistical difference in the remaining data.

#### 3.3.3. The Effectiveness of the Ti Plates in the Fracture Fixation

As shown in Figure 6, in the 24th week, both the Synthes-Ti plate and 3D-Ti plate groups showed complete fracture healing: the bone surface was smooth, and the fracture line was invisible. Even in the 4th week, the fracture line was difficult to discern. Additionally, all plates were kept in place at all times.

Figure 7a shows the parts of fracture healing after CT reconstruction, as well as the fact that there were no differences between the two groups in the 4th or the 24th week. Combined with the results shown in Figure 7b, it can be seen that the BV/TV, Tb. N, and Tb. Th of the two groups increased, and the Tb. Sp decreased in the 24th week in comparison with the 4th week. In the 24th week, the BV/TV, Tb. N, Tb. Th, and Tb. Sp of the Synthes-Ti plates were all higher than the 3D-Ti plates, but there was no statistical difference between the two groups (*p* > 0.05, *p* > 0.05, *p* > 0.05, *p* > 0.05, separately).

In Figure 8, the representative H&E staining images presented the condition of fracture healing in the 4th and 24th weeks. The images showed normal osteocytes at the fracture sites in both groups, without prominent pink-colored cartilaginous calluses or fibrous tissue. Masson staining revealed the changes in osteocytes. The amount of new bone stained red was higher in the 4th week when the fracture was still healing. However, in the 24th week, almost all the bone tissue was stained blue with little red, indicating that the fracture had healed (Figure 8).

#### 3.3.4. Mechanical Stability of 3D-Ti Plate

After 24 weeks of implantation, the elasticity modulus of the 3D-Ti plates was still more robust than that of the control group and maintained the same strength as the original 3D-Ti plates. However, The ultimate stress of the implanted Ti plates of the two groups was lower than the original plates (*p* < 0.05) (Figure 6), illustrating that the Ti plates became more easily deformed after the long-term influence of the internal environment.

## 4. Discussion

Ti alloy plates perform a crucial function in rigid internal fixation in maxillofacial surgery [1,2]. Although 3D-printed Ti alloy plates have long been used in clinical practice [10,11], previous studies still need a comprehensive evaluation of them [2,12,13]. Therefore, the present study aimed to evaluate the mechanical properties, surface morphology, biocompatibility, and application accuracy of the 3D-Ti plates by comparing them with the Synthes-Ti plates.

The study used Ti alloy Ti6Al4V powder to manufacture the 3D-Ti plates through SLM. Heavy metal ions, including titanium and vanadium ions, can cause cell membrane rupture and organelle damage and lead to cell necrosis [14]. Xuezhi Lin et al. showed that SLM triggers tight metal ion binding with a narrower metal ion gap, a more minor degree of the precipitation of heavy metal ions, and therefore no toxic side effects on cells [15]. Firstly, from the SEM results of the 3D-Ti plate surface, no granular metal powder was observed, even at high magnification. Furthermore, we performed both *in vitro* and *in vivo* experiments, which simulated implantation into the internal environment. The results showed good biocompatibility of the two Ti plates in the current study. Experiments in rabbit models concluded that the biosafety of 3D-Ti plates is comparable to that of conventionally fabricated Ti plates [16].

Surface roughness can directly affect the Ti plate’s biological properties by changing the cell behavior [17]. Some previous researchers have studied the surface roughness of the Ti alloy plates. Ponader et al. found that the cells could enter the proliferation stage on the surfaces, with the Sa value not exceeding 24.9 μm. When the Ra value is more significant than 56.9 μm, it is not conducive to cell proliferation [18]. According to the in vitro experiment results, the surface roughness of the Ti plates was below 24.9 mm, which made BMSCs adhere and grow well. Another way to interpret it is that 0% to 25.8% of the patients who undergo orthognathic surgery require the removal of at least one plate, and 0% to 18.3% of plates require removal [19]. The roughness of the 3D-Ti plates was 106.44 nm, and that of the Synthes-Ti plates was 209.46 nm, far below the above value, and the low surface roughness reduces the attachment to nearby tissues, which is more conducive to the removal of plates later. However, Mazurek-Popczyk obtained an Sa of 131 nm for 3D-Ti plates and 60.67 nm for Synthes-Ti plates, which was dissimilar from our results [20]. We believe this is reasonable, considering such factors as processing techniques and testing parameters.

Ti alloy plates are widely used in maxillofacial surgery. The direct rigid internal fixation of the fracture provides a stable and safe environment for healing and can prevent the displacement of the broken end during functional movement [21]. Complications, some of which may be associated with functional movements, still exist with the fixation that may either lead to the loss or fracture of a plate, which is occasionally reported in clinical practice [22]. Therefore, the biomechanical properties of the Ti plates exert a vital role in fracture healing as they provide sufficient strength to reduce the movements between the bone segments during fracture healing [12]. The thickness of 3D-Ti plates for fracture fixation should be 1 mm [10]. At the same thickness, the mechanical properties of 3D-Ti plates were significantly better than those of standard Ti plates [13]. In this experiment, two Ti plates with a thickness of 1 mm were used to obtain the same experimental results. In addition, by comparing the mechanical properties between the original and implanted Ti plates, the 3D-Ti plates in this study were able to achieve good rigid internal fixation and keep the mechanical strength unchanged, providing a stable environment for fracture healing.

Unfortunately, due to the maxillofacial region’s irregular shape and complex anatomical structure, the standard Ti plates were often supposed to be shaped to adapt to the bone surface during fixation [2,3]. Hong and Liu mentioned that the bone–plate contact rate of the standard Ti plate was only 53% [2,3], which could be increased to 66% when the Ti plate was manually bent in this study. Lauria et al. found by designing cycle tests that the manual bending of the Ti plates may cause microfractures and brittle points, which leads to the premature fracture of the shaped Ti plates [11]. These complications can be effectively avoided by using the 3D-Ti plates [5,11]. This study verified that the 3D-Ti plates could fit the bone surface well and avoid bending the Ti plate on the rabbit head model.

Following the recent development of 3D-printing technology, there have been several reports on the application of individualized Ti plates in craniomaxillofacial surgery [23,24], such as in the field of oncology [25,26,27]. The evidence suggests that using 3D-Ti plates enables the surgeon to accurately transfer the virtual surgical plan to the patient [28]. Therefore, using customized 3D-Ti plates in orthognathic surgery has great potential [29]. In 2015, Mazzoni et al. published the first paper dealing with this technique’s accuracy [30]. They assessed the accuracy and concluded that the customized Ti plates achieved a reproducibility of <2 mm for 100% of the facial surface in 7 out of 10 cases. Heufelder et al. [31] and Li et al. [32] assessed the accuracy by measuring positional differences between planned and postoperative landmarks. The maximum absolute positional differences among cases for the maxilla were 2.02 mm and 0.74 mm, respectively. The total error rate of maxillary repositioning was 0.62 mm between the virtual plan and the postoperative result [33]. These comparative studies with a limited number of cases suggested that there is always a margin of error in the actual procedure where the virtual surgical protocol is transferred to the patient. From this experiment, the bone–plate contact rate of 3D-Ti plates was up to 98.1%, which could verify the accuracy of 3D-Ti plates to a certain extent. In the future, we will carry out many randomized, controlled trials to explore the accuracy of the application of 3D-Ti plates from virtual surgery to actual surgery.

## 5. Conclusions

Following a combined *in vivo* and *in vitro* study, the overall results revealed shallow and non-significant differences regarding biocompatibility between the 3D-Ti plates and the Synthes-Ti plates. However, the 3D-Ti plate shows superior mechanical properties and accuracy. In addition, 3D-Ti plates represent a promising alternative to the standard Ti plates.

## Figures and Tables

**Figure 1 jcm-12-00444-f001:**
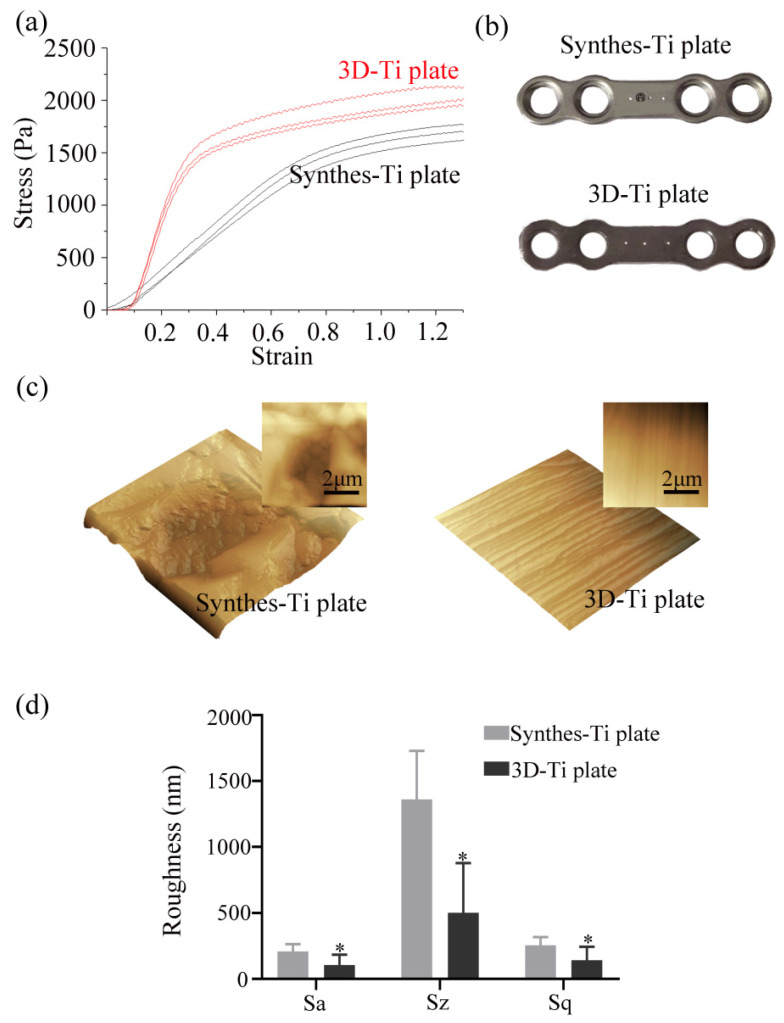
Mechanical property and surface morphology of the two groups of Ti plates. (**a**). The stress-strain curves of the two groups of Ti plates. The red curves were the 3D-Ti plates, and the black curves were the Synthes-Ti plates. (**b**) Indentations on the surface of the plates under 10 kg of force. (**c**) AFM images of the Ti plates surfaces. The plates were directly scanned (top right) and reconstructed. (**d**) Quantitative analysis of surface roughness by AFM. Sa, average surface roughness; Sz, vertical range surface roughness; Sq, rootmean-square surface roughness. * *p* < 0.05.

**Figure 2 jcm-12-00444-f002:**
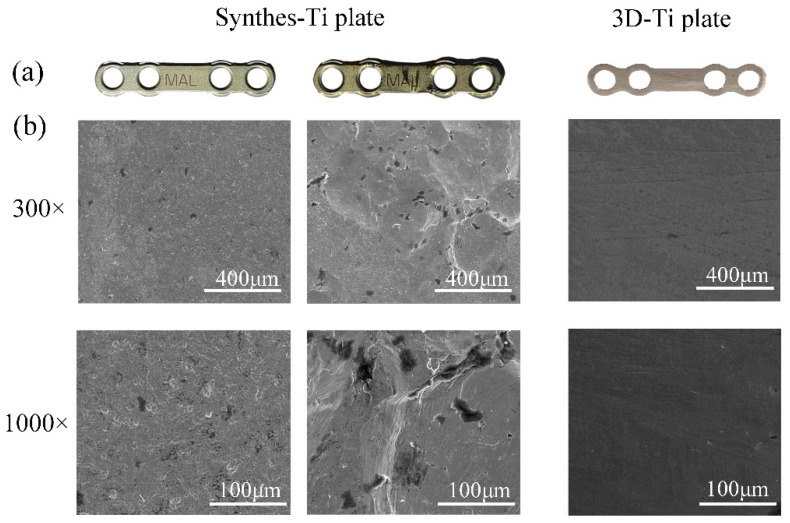
Macroscopic and microscopic surfaces of the two groups of Ti plates. (**a**) Images of the macroscopic surfaces of the Ti plates. (**b**) SEM images of the surface morphology of Ti plates at different magnifications. The left two columns are the Synthes-Ti plates, the original one on the left and the shaped one on the right. The rightest column is the 3D-Ti plate.

**Figure 3 jcm-12-00444-f003:**
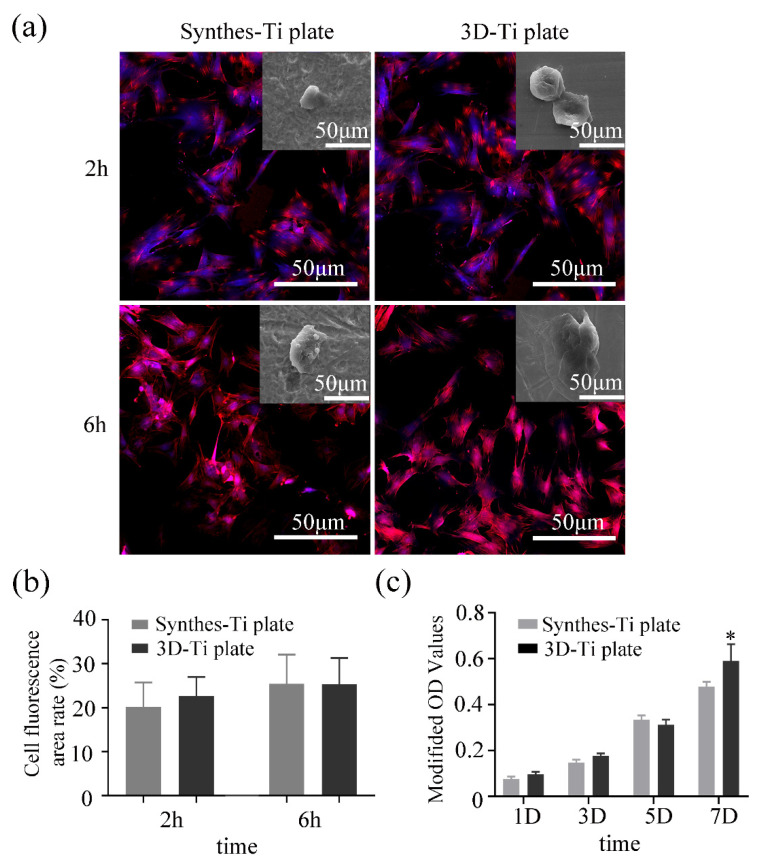
Cytocompatibility evaluation of the two groups of Ti plates. (**a**) HBMSCs morphology at the 2 h and 6 h incubation on the surfaces of the Ti plates investigated by CLSM and SEM (inserted). The cytoskeleton was fluorescently labeled in red. (**b**) Quantitative analysis of cell fluorescence area. (**c**) Quantitative analysis of BMSCs proliferation cultured on the Ti plates by cck-8 assay. * *p* < 0.05.

**Figure 4 jcm-12-00444-f004:**
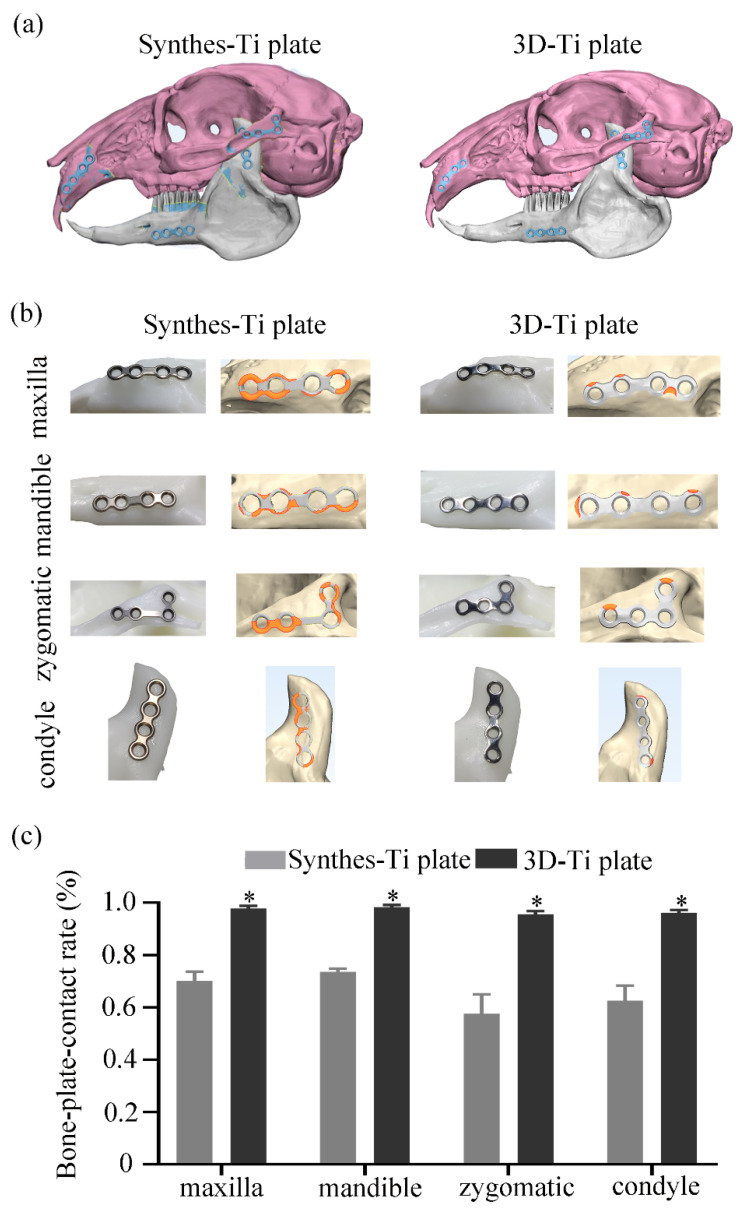
Evaluation of the accuracy of the plate shaping between the two groups of Ti plates. (**a**) Digital models of the rabbit head and mounted Ti plates on the specific surfaces. (**b**) Representative magnified images of the plates and the bone surfaces of the two groups. In each group, the left column are the images of the Ti plates at each position on the resin models, and the right column shows the contact between the plates and the bone surfaces in 3-Matic software. The orange area marks the part of the Ti plate that is not in contact with the bone surface. (**c**) Quantitative analysis of bone-plate contact rate of the Ti plates at each position. * *p* < 0.05.

**Figure 5 jcm-12-00444-f005:**
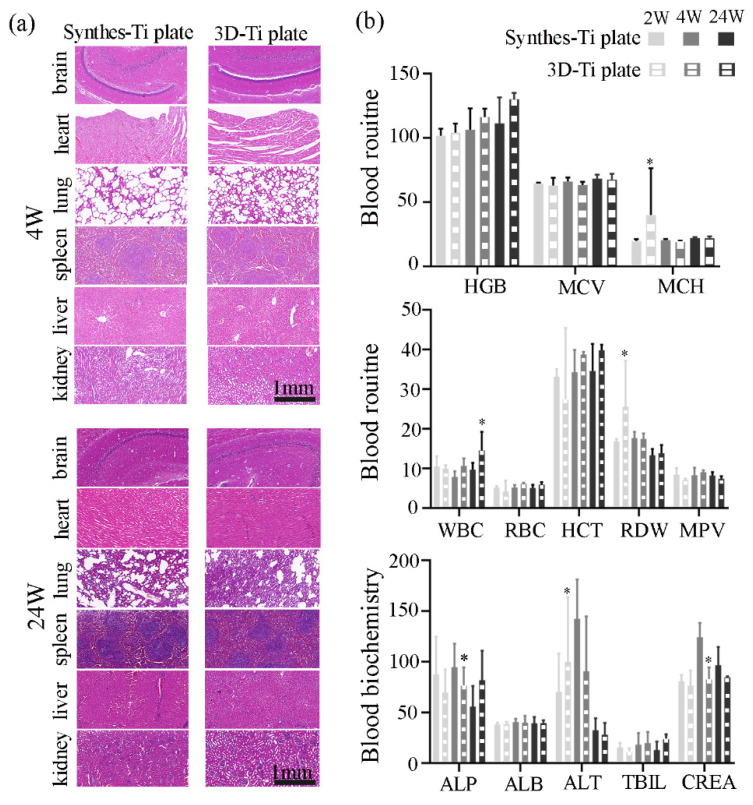
Systemic safety and biocompatibility evaluation of the implanted Ti plates between the two groups. (a) Histological examination of different visceral tissues through H&E staining 4 and 24 weeks after surgery. Bar = 1 mm. (b) Quantitative analysis of blood routine and blood biochemistry 2, 4 and 24 weeks after surgery. HGB, hemoglobin concentration (g/L); MCV, mean corpuscular volume (fL); MCH, mean red blood cell hemoglobin volume (pg). WBC, white blood cell counts (109/L); RBC, red blood cell counts (1012/L); HCT, hematocrit (%); RDW, red blood cell distribution width (%); MPV, mean platelet volume (fL). ALT, alanine aminotransferase (U/L); ALB, albumin (g/L); ALP, alkaline phosphatase (U/L); TBIL, total bilirubin (μmol/L); CREA, creatinine (μmol/L). * *p* < 0.05.

**Figure 6 jcm-12-00444-f006:**
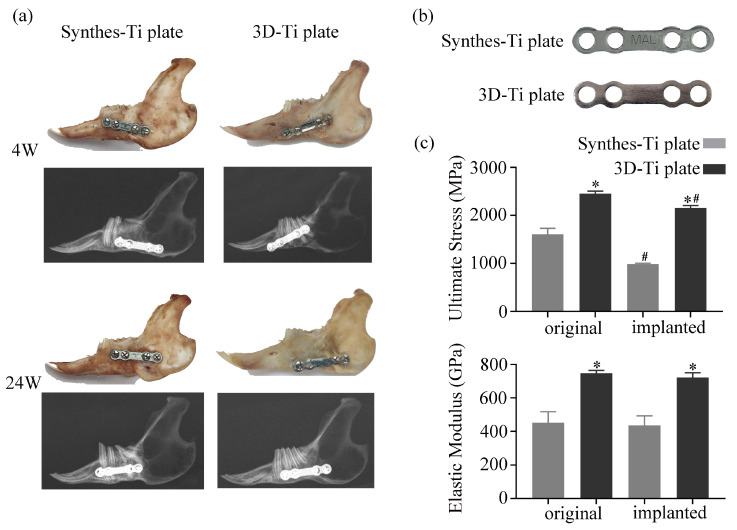
Evaluation of fixation and mechanical stability between the two groups of Ti plates. (**a**) Gross appearance (upper) and X-ray images (lower) of the mandible implanted different Ti plates 4 and 24 weeks after surgery. (**b**) Representative images of the removed Ti plates 24 weeks after surgery. (**c**) Comparison of the mechanical stability between the two groups of Ti plates. * *p* < 0.05 vs. Synthes-Ti plate, # *p* < 0.05 vs. original plate.

**Figure 7 jcm-12-00444-f007:**
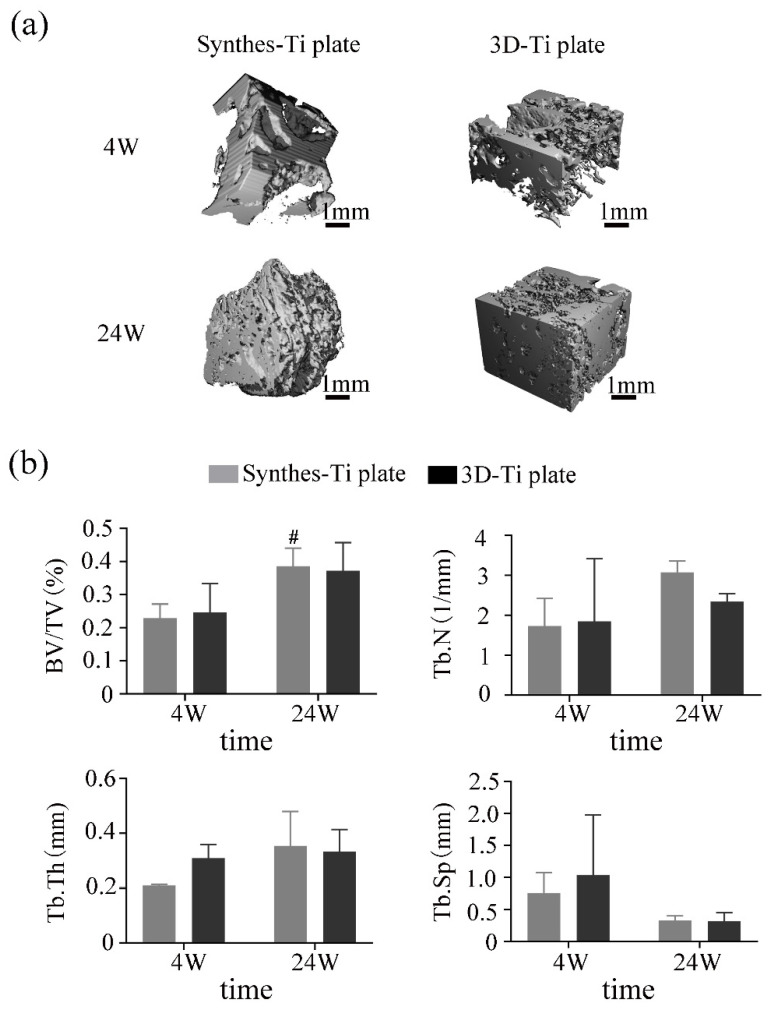
CT evaluation of fracture healing between the two groups. (**a**) Representative Micro-CT reconstruction images of the area of fracture healing. (**b**) Quantitative analysis of fracture healing 24 weeks after surgery. BV, bone volume; TV, total volume; Tb.N, trabecular number; Tb.Th, trabecular thickness; Tb.Sp, trabecular separation. # *p* < 0.05 vs. 4W.

**Figure 8 jcm-12-00444-f008:**
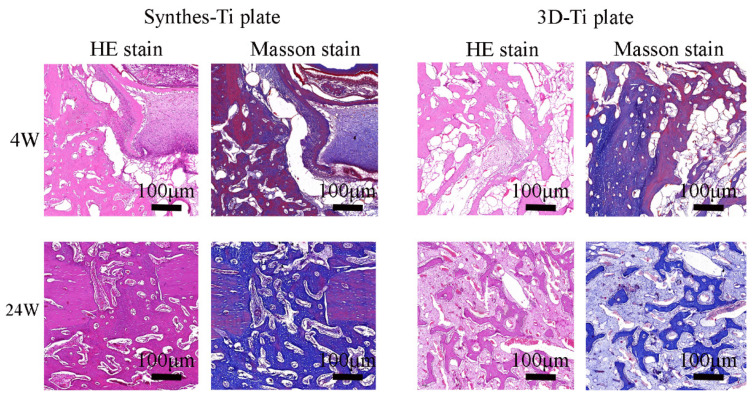
Histological evaluation of fracture healing between the two groups.

**Table 1 jcm-12-00444-t001:** Comparison of the mechanical test results of the two groups of plates.

	Hardness (HV10)	Ultimate Stress (MPa)	Elasticity Modulus (GPa)
Synthes-Ti plate	156.04 ± 1.37	1670.77 ± 52.81	451.75 ± 66.139
3D-Ti plate	370.78 ± 1.25	2153.13 ± 62.15	747.81 ± 16.813
*p*	<0.0001	<0.0001	<0.0001

Sample *t*-test. Data are expressed as Mean ± SD (*n* = 3).

## Data Availability

The data presented in this study are available on request from the corresponding author. The data are not publicly available due to privacy.

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
