# Peer review of "Evaluation of the Properties of 3D-Printed Ti Alloy Plates: In Vivo and In Vitro Comparative Experimental Study"

_jcm, 2023, doi:10.3390/jcm12020444_

Round 1

Reviewer 1 Report

In this manuscript, the authors show ‘Evaluation of the properties of the 3D printed Ti alloy plates: In vivo and in vitro comparative experimental study’. They present a nice spectrum of microstructure, mechanical and biological performance. The results are very interesting and the paper is generally well-prepared. However, in my opinion, there are some points that authors should change to improve the paper's quality. During its careful review, a series of problems and suggestions (listed below) arose.

My recommendation is: Minor revision.

Comments

1.      The highlights and abstract should be re-write. What is the variance parameter? The quantitative results are required in highlights and abstract.

2.      In section 2.4, please mention the dimension of samples and room temperature in three-point bending test conditions and standards.

3.      Hardness measurements: please add images of indentations. Such images will increase the value of the manuscript.

4.      English expression needs to be improved. Does the manuscript contain some grammatical errors and unreasonable sentence structure?

Author Response

Thank you for your letter and the reviewers’ comments concerning our manuscript entitled “Evaluation of the properties of the 3D printed Ti alloy plates: In vivo and in vitro comparative experimental study ” (jcm-2128414). Those comments are valuable and very helpful. We have read through comments carefully and have made corrections. Based on the instructions provided in your letter, we uploaded the file of the revised manuscript. We have used the “Track Changes” function to make up all the revisions. The responses to the reviewer's comments are marked in red and presented following.

We would love to thank you for allowing us to resubmit a revised copy of the manuscript and we highly appreciate your time and consideration.

Point 1: The highlights and abstract should be re-write. What is the variance parameter? The quantitative results are required in highlights and abstract.

 Response 1: Thank you for your comments. We have re-written the abstract and added the quantitative results.

Point 2: In section 2.4, please mention the dimension of samples and room temperature in three-point bending test conditions and standards.

Response 2: Based on your suggestion, we have added this into the section 2.4.

 Point 3: Hardness measurements: please add images of indentations. Such images will increase the value of the manuscript.

Response 3: Based on your suggestion, we have added this into the figure 1.

 Point 4: English expression needs to be improved. Does the manuscript contain some grammatical errors and unreasonable sentence structure?

Response 4: We apologize for the language problems in the original manuscript. The language presentation was improved with assistance from a native English speaker with appropriate research background.

Reviewer 2 Report

The authors have performed an impressive study on the properties of 3D printed plates. The study was well designed and executed. The sample size of 24 animals was appropriate to obtain a significant power of the results.

The design by comparing 3D printed plates directly with stock plates in vivo was well thought out. I especially liked the comparison if changes in surface structure from stock plates compared with after manual adaptation of stock plates. This was clinically relevant and interesting. Furthermore, the comparison in ultimate load and ultimate stress between the plates and before and after implantation of 4 or 24 weeks is very interesting and well designed. I really enjoyed reading these results in particular as this design detail is also very interesting. The authors chose to combine the 4-week and 24-week cohort, which is acceptable as the cohort did not seem to differ between 4 and 24 weeks.

The only thing not included in this study is the fatigue of the materials measured using repeated loadning of the plates. I hope that the authors plan on performing a study including this parameter, as this has not been performed to the best of my knowledge. This is an area that can be of concern regarding the clinical application of 3D printed plates as this parameter more accurately simulate the clinical conditions that apply to fracture and osteotomy healing. This aspect is not necessary for this publication but simply a suggestion for future studies. The authors incorporated some of this aspect in the study design as they performed osteotomies to simulate fracture of the mandible and no plates were reported lost or fractured.

I wish to thank the authors for an outstanding work and for their hard work and dedication to advancing the clinical application in 3D printing. 

Author Response

Thank you very much for your comments. Fatigue performance is an important indicator. The lack of experimental design about the fatigue of the Ti plates is a pity. Due to the small size of our experimental Ti plates, it is difficult to perform this test on a large fatigue test instrument. We are actively planning on performing a study including this parameter. 
